# The First Early Evidence of the Privacy-Invasive Use of Browser Fingerprinting for Online Tracking

## ABSTRACT

While advertising has become commonplace in today's online interactions, there is a notable dearth of research investigating the extent to which browser fingerprinting is harnessed for user tracking and targeted advertising. Prior studies only measured whether fingerprinting-related scripts are being run on the websites but that in itself *does not* necessarily mean that fingerprinting is being used for the privacy-invasive purpose of online tracking because fingerprinting might be deployed for the defensive purposes of bot/fraud detection and user authentication. It is imperative to address the mounting concerns regarding the utilization of browser fingerprinting in the realm of online advertising.

To understand the privacy-invasive use of fingerprinting for user tracking, this paper introduces a new framework "FPTrace" (fingerprinting-based tracking assessment and comprehensive evaluation framework) designed to identify alterations in advertisements resulting from adjustments in browser fingerprinting settings. Our approach involves emulating genuine user interactions, capturing advertiser bid data, and closely monitoring HTTP information. Using FPTrace, we conduct a large scale measurement study to identify whether browser fingerprinting is being used for the purpose of user tracking and ad targeting. The results we have obtained provide robust evidence supporting the utilization of browser fingerprinting for the purposes of advertisement tracking and targeting. This is substantiated by significant disparities in bid values and a reduction in HTTP records subsequent to changes in fingerprinting. We additionally demonstrate the potential use of fingerprinting for privacy-evading online tracking purposes even when users opt out of tracking under GDPR/CCPA regulations. In conclusion, our research unveils the widespread employment of browser fingerprinting in online advertising, prompting critical considerations regarding user privacy and data security within the digital advertising landscape.

## 1 INTRODUCTION

Browser fingerprinting is a technique employed to surreptitiously collect data regarding a user's web browser settings during their online activities. The collected data is then utilized to construct a unique digital identity, commonly referred to as a 'fingerprint,' for that specific user browser. Each time a user visits a website, there is potential for the site to employ browser fingerprinting as a means to identify or track the user. Many earlier research studies, such as Englehardt et al. [24] and Lerner et al. [31] and reports such as [1], *assumed that the adoption of a fingerprinting script itself is an indication of web tracking and a violation of web privacy*. However, this assumption *does not* hold — just like cookies, browser fingerprinting can be used for defensive security purposes, like bot/fraud detection or authentication. For example, Wu et al. [45] show that adversarial fingerprints are different from those of benign users and therefore many real-world websites are using fingerprints for bot/fraud detection. As another example, Lin et al. [32] have demonstrated the real-world usage of browser fingerprinting in

authentication just like what Laperdrix et al. [28] have illustrated in a feasibility study.

Therefore, the research question that we are answering in this paper is: *whether browser fingerprints are indeed adopted for online tracking, thus violating web privacy*. To the best of our knowledge, none of the prior works have established the link between browser fingerprinting and online tracking. On one hand, many works [24, 31], as mentioned above, consider the existence of fingerprinting scripts as a means of online tracking, which is not true. On the other hand, people have studied the relationship between personalized advertisements and web tracking in general, like cookie-based tracking. For instance, Wills et al. [44] explored ad tracking on the Google and Facebook advertising platforms. Similarly, Zeng et al. [47] employed header bidding to assess targeted ads. These studies did not specifically address the methods employed to link tracking with online advertising; therefore, it remains unclear whether browser fingerprinting contributes to online tracking and privacy violation.

This paper seeks to bridge this gap in current research and regulatory assessment practices by investigating whether advertising ecosystem indeed utilizes browser fingerprinting for user tracking and targeting via a measurement study. Our *key* insight is that if browser fingerprinting plays a role in online tracking, the change of fingerprints will also affect the bidding of advertising and the underlying HTTP records. Specifically, our approach involves leaking user interest data through controlled A/B experiments, modifying browser fingerprints, and leveraging advertiser bidding behavior and HTTP events as a contextual indicator in the advertising ecosystem to deduce changes in advertisements. Given that advertiser bidding behavior and HTTP events are influenced by their prior knowledge of the user, we anticipate notable changes in this information when altering browser fingerprints.

**Our Contributions:** We offer the first study to measure whether browser fingerprinting is being used for the privacy-invasive purposes of user tracking, targeting and advertising. Our main contributions can be summarized as follows:

(1) We introduce a framework, FPTrace, for detecting changes in advertisements following alterations in browser fingerprinting. FPTrace simulates real user interactions, captures advertiser bids, records HTTP data, and removes or exports cookies to observe such changes for the measurement of purposes of browser fingerprints.

(2) Our findings provide evidence that browser fingerprinting is indeed utilized in advertisement tracking and targeting. The bid value dataset exhibits notable differences in trends, mean values, median values, and maximum values after changing browser fingerprints. Moreover, the number of HTTP records, encompassing HTTP chains and syncing events, decreases significantly after altering browser fingerprints. We also evaluate the role of browser fingerprinting in cookie restoration. Our results confirm that certain cookies contain browser fingerprinting information. We documented 378 instances of cookie restoration related to

fingerprinting across 90 unique combinations of cookie keys and host pairs across all settings. However, there is no conclusive evidence to support browser fingerprinting's direct involvement in cookie restoration after we did the manual inspection.

(3) We further study the potential malicious use of fingerprinting in the presence of data protection regulations such as GDPR and CCPA when used with content management platforms. Even under the GDPR and CCPA regulation protections, there are significant variations in the number of HTTP chains and syncing events observed in certain instances when browser fingerprints are altered. Under GDPR, websites utilizing Onetrust, Quantcast, and NAI might be involved in data sharing activities that use browser fingerprinting to identify users. Under CCPA, Onetrust, and NAI might be involved in data sharing activities that use browser fingerprinting to identify users.

## 2 BACKGROUND

### 2.1 Browser Fingerprinting

Browser fingerprinting is a technique to identify users using a bunch of data collected from users' browsers, which may be unique to each browser (we refer to [29] for a representative survey in browser fingerprinting).

Yen et al. [46] and Nikiforakis et al. [39] assess the efficacy of fingerprinting techniques. Vastel et al. [43] examine the discrepancies present in browser fingerprints. Boda et al. [19] and Cao et al. [20] investigate alternative aspects of browser fingerprinting. Previous studies have explored various facets of browser fingerprinting, including techniques involving canvas [36], JavaScript [37], and hardware [38] methods. Information like User Agent which contains the users' browser version and operating system version can be obtained by the website via HTTP header. Screen resolution, canvas fingerprints, users' current time zone information can be accessed by the website via JavaScript APIs. Such information can be gathered and used to construct a user profile. Eckersley [23], Fifield et al. [25] and Laperdrix et al. [30] have shown that the browser fingerprints can be used for user identification due to their uniqueness. Acar et al. [18] and Iqbal et al. [27] demonstrated that browser fingerprints can be utilized in fraud detection. In real-world applications, services like Datadome [4], Radware [5], and HUMAN [12] employ browser fingerprinting for the purpose of fraud detection. Previous research such as [41] and articles like [1] solely assessed the presence of fingerprinting-related scripts on websites. However, this alone does not indicate malicious intent for user tracking.

Website "COVER YOUR TRACK" can display each browser fingerprint feature [3]. Website "AM I UNIQUE" can also show the uniqueness of each browser fingerprint among all other users [9]. Website "FingerprintJS" generate a unique ID for each user by using the browser fingerprints, and also display the ID to user [17].

### 2.2 Browser Fingerprint Spoofing

Spoofing browser fingerprints enables the concurrent operation of multiple browser instances on a single device, each with distinct browser fingerprinting. The Gummy Browser poses a significant threat to the privacy and security of applications that rely on browser fingerprinting [34]. This tool employs three distinct methods of attack, script injection, script modification, and the exploitation of the browser's inherent settings and debugging tools.

Notably, the Gummy Browser exhibits the capability to successfully emulate all JavaScript-based browser fingerprinting techniques, including the sophisticated canvas fingerprinting method. In the context of script modification, the attacker wields the ability to manipulate the behavior of the JavaScript API "toDataURL()" by substituting it with either a predefined or randomized string. Conversely, in the case of script injection, the attacker can strategically introduce a breakpoint within the script precisely at the juncture where the JavaScript API "toDataURL()" is invoked, thereby enabling the replacement of its value with either a predetermined or random string.

### 2.3 Cookie Restoration

Cookies play a crucial role in recognizing devices, such as computers, within a network. Some cookies are designed to pinpoint individual users, enhancing their web browsing experience. When these cookies are deleted or expire, cookie restoration becomes valuable for reinstating them. Users have the option to safeguard their cookies for potential restoration by employing extensions that export cookies to JSON files or by manually copying cookie data from their browser profiles in case of accidental deletion. On the other hand, websites may endeavor to retain cookies within users' browsers and restore them even after users have deleted the cookies, thereby maintaining user tracking capabilities.

### 2.4 Header Bidding

Header bidding [6] is a method employed by publishers on websites. Here, publishers designate specific advertising spaces for potential advertisers. The advertiser securing the highest bid gains the chance to display their ads in the corresponding slots. In client-side header bidding, users have the convenience of directly accessing and observing all the bids from their web browsers. Prebid.js [11] is a notable implementation of header bidding. Through the API *pbjs.getBidResponses()*, users on the client side can inspect the list of advertisers who engaged in the bidding process to secure the opportunity to display ads during the current user's visit. In the study outlined in [40], the author observes that profiles classified as "Only category" command prices around 40% higher than those assigned to "New user" profiles. The key finding underscores that advertisers' bidding behavior is shaped by their prior familiarity with the user, resulting in elevated bid values compared to users for whom advertisers lack previous knowledge. Liu et al. [33] additionally demonstrated that advertisers with knowledge of users through data syncing tend to submit higher bid values in header bidding.

### 2.5 Privacy Regulations

Recently, the European Union introduced the General Data Protection Regulation (GDPR) [15], and California introduced the California Consumer Privacy Act (CCPA) [16], both possessing the potential to regulate and restrain the online advertising and tracking ecosystem. GDPR requires that online services secure user consent (Articles 4 (11)) prior to processing user data (Article 6 (1) (a)). CCPA requires online services to offer users the ability to opt-out of the sale of their personal data (Section 1798 (a) (1)).

Both GDPR and CCPA require that websites must provide privacy notices containing information and controls for opting in/out of the collection and processing of personal information, which should include browser fingerprints. To comply with these requirements, websites are required to integrate consent management platforms (CMPs) [14]. CMPs scan websites, identifying all cookies set by HTTP headers and scripts, no matter first or third-party resources. In the case of GDPR, CMPs must ensure that only necessary cookies are shared, and consent is obtained before sharing non-essential cookies. In the case of CCPA, CMPs should ensure that they provide users with controls to opt-out of the sale of personal information.

## 3 FPTRACE: OUR FRAMEWORK TO ASSESS FINGERPRINTING USE IN TRACKING

### 3.1 Framework Design

The objective of this research is to explore the utilization of finger-printing in the realms of cookie restoration and targeted advertising. To achieve this, it is imperative to design a robust framework tailored for efficient data acquisition. Given the limitations of manual data collection, our proposed framework "FPTrace" is engineered to automate this process.

FPTrace is capable of emulating human user behavior by systematically visiting a curated list of websites categorized under specific keywords, such as "Computer". During the visit, FPTrace not only loads the website but also simulates the behavior of a real human user by browsing the webpage. It is capable of clicking on various links or advertisements, moving the mouse to specific areas on the webpage, and even interacting with functions provided by third-party services, such as Consent Management Platforms (CMPs) or standard cookie banners. After exploring the websites, FPTrace is able to compile a browser profile, encompassing details like browsing history and cookie data. This comprehensive profile is designated as the "Interest Persona", serving as a representation of user interests and preferences. Furthermore, it has the capability to extract data directly from advertisers, facilitating an in-depth analysis of targeted advertising triggers. FPTrace is capable of capturing and recording data such as bidding behaviors from various advertisers, as well as personal data syncing events. This information is then securely stored within the local database of FPTrace.

The design ensures versatility by allowing the FPTrace to sequentially or concurrently access multiple websites. Additionally, FPTrace is equipped to export and compare cookies after a stipulated number of website visits. To enhance its relevance in the domain of fingerprinting, integrated features have been incorporated for browser fingerprinting spoofing.

### 3.2 Framework Implementation

We present the tools we utilized in the FPTrace framework and the functions we implemented to meet the varying requirements of our experiments.

*3.2.1 Web Crawling.* We used OpenWPM [24] to construct our crawling FPTrace. OpenWPM is a widely used tool to collect large amount of data related to privacy. OpenWPM relied on the Selenium [13] to handle browser Firefox to do automated crawling.

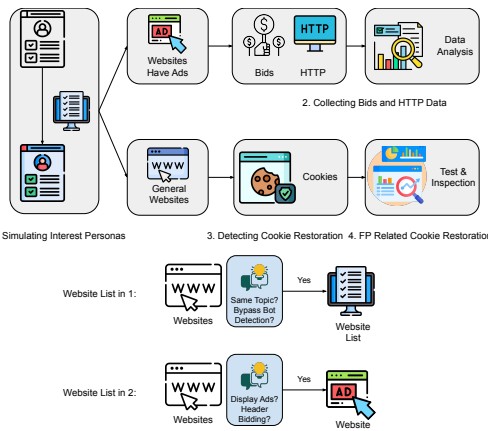

***Figure 1:*** *High level overview of measurement study methodology. In **Step 1**, We create browser persona by visiting a list of websites. This step is called "Simulating Interest Personas". In **Step 2**, we first use trained persona to visit websites which display ads, then collect bids and HTTP data. This step is called "Collecting Bids and HTTP Data". In **Step 3**, we extract cookies from the browser profile and compare them between different experiment settings. This step is called "Detecting Cookie Restoration". In **Step 4**, we analyze the manual inspect extracted cookies. This step is called "Detecting Fingerprinting Related Cookie Restoration".*

OpenWPM has default functionality for browsing a list of websites, saving and loading browser profiles, and recording HTTP data. However, our experiments require further parameters. We added functionality to the OpenWPM to gather bid information.

For collecting bids, we used the Prebid.js, a popular implementation of the header bidding protocol [11], mostly because it takes place on the client side, enabling us to intercept the bidding process [2]. The bids objects we get from Prebid.js include all partipated advertisers information, and the ads image HTML contents.

We have incorporated a feature that allows the removal of all cookies from the browser profile. This function ensures that the process of data training and collection is completed automatically, free from any influence by human behaviors.

While gathering data through web crawling, we integrated functionality to mimic typical human behaviors. FPTrace enables the simulation of mouse movement from the top to the bottom of web pages, accompanied by scrolling down, and introduces random waiting periods ranging from 1 to 10 seconds. This feature enhances the realism of the crawler's interaction with the website, resembling genuine human browsing behavior.

*3.2.2 Spoofing Browser Fingerprints.* Our experiments require the use of different browser fingerprints. We implemented an extension to spoof JavaScript based browser fingerprints features.

By spoofing JavaScript APIs, we used real browser fingerprints data from an dataset on Github [42]. The data is collected from real devices. In spoofing JavaScript APIs, we used Script Injection code provided by Gummy Browser [35] to overwrite all possible JavaScript APIs that can be used in browser fingerprints, such as Navigator(API: *Navigator*), Screen(API: *Screen*), Canvas(API: *canvas.getContext*) and Date(API: *Date*) APIs. To ensure the spoofing of all APIs prior

to the loading of any webpage, we encapsulated all spoofing scripts within a browser extension.

Only spoofing JavaScript APIs may not work if websites match the value from JavaScripts to the one from HTTP header. So we have also spoofed the HTTP header. ModHeader [8] is an extension to allow users to modify HTTP headers. We embedded Selenium ModHeader [7] version into OpenWPM to spoof HTTP header.

*3.2.3 Capturing Bidding Object.* Header bidding can signal the enthusiasm of potential bidders. A bidder interested in showcasing their advertisements in the present slot will typically place higher bids. Cook et al. [22] have proved that the header bidding can be used to identify if the user is tracked. A higher bid can represent the user is tracked, and the user information is used by publishers, bidders, or bidding auctions. Specifically, we use Prebid.js to monitor bidders' behaviors. Prebid.js is an implementation of header bidding protocol, and the user can get bidders' behaviors, including the winner of each ad slot, the ads image HTML contents, and also all of the participated bidders bids value no matter the bidders win or lose.

OpenWPM does not have functions to record bidding behaviors, so we inject codes to record those behaviors when the crawlers browse the website with Prebid.js and displaying ads.

*3.2.4 Exporting Cookies.* Grasping the significance of browser fingerprints in advertising also necessitates an examination of cookie data. Fingerprints could become crucial if cookies that were deleted get reinstated. OpenWPM currently lacks functionality for exporting complete cookie data after each website visit by the crawler, or after visiting a random selection of websites. Therefore, we have developed a function that allows for the export of all cookies, both first and third-party, accumulated in the browser managed by OpenWPM, regardless of the circumstances.

## 4 MEASUREMENT STUDY METHODOLOGY

In this section, our measurement study methodology that utilizes the FPTrace framework is introduced. Initially, in Section 4.1, we outline the simulation of interest personas, which emulate real person's interests. Following this, in Section 4.2, the collection of bids and HTTP data using various settings is presented, employing simulated interest personas to visit websites that use Prebid.js. Subsequently, the detection of cookie restoration post removal is discussed in Section 4.3, followed by an elaboration on the identification of browser fingerprinting-associated cookie restoration in Section 4.4. Figure 1 provides an overview of the measurement study methodology. Finally, in Appendix A.3, the configuration for discerning whether browser fingerprinting persists for data collection post-user opt-out under GDPR and CCPA protection is discussed.

### 4.1 Simulating Interest Personas

A Persona encapsulates key attributes of a real individual, encompassing aspects such as personal interests, characteristics, and traits. Likewise, Browser Personas serve to embody the browsing preferences of web users, encompassing factors like desired purchases, vacation destinations, and preferred news topics. Websites and third-party services seek to understand, track, and tailor personal advertisements to users, making Browser Interest Personas invaluable in this endeavor. To understand how advertisers perform bidding based on

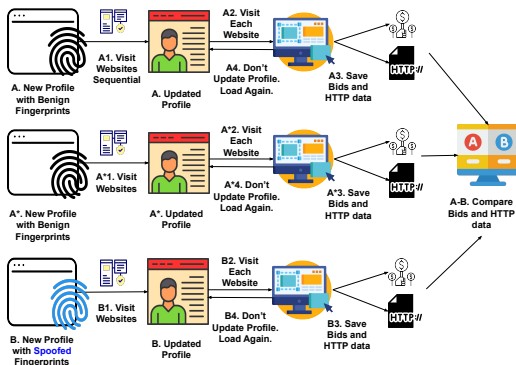

*Figure 2: High level overview of advertisement experiment. In **step 1**, FPTrace will visit websites sequentially to keep updating the browser profile. In **Step 2**, FPTrace will control the browser with updated profile to visit each website. In **Step 3**, FPTrace will record the bids data and HTTP data. In **Step 4**, FPTrace will not update the current browser profile, and load the profile updated after **Step 1**. After visiting all websites in **Step 2**, FPTrace will compare the bids data and HTTP data between **A**, **A\*** and **B**.*

users' interests, it is essential to simulate browsers with specific interests before engaging in Prebid.js auctions for displaying ads. We begin by compiling a list of websites, denoted as $W_p$, within a particular topic. Alexa provides topic-specific website lists, but not all are closely related to the topic keywords. To address this, we employ Google search to construct distinct website lists. We select the first 40 websites from the search results and manually inspect them to eliminate duplicates.

Once we have compiled the website list $W_p$, we need to bypass web driver detection, as OpenWPM uses Selenium for automation. This involves overwriting the JavaScript API Navigator.webdriver, which distinguishes between real user visits and automation.

Subsequently, we direct OpenWPM to sequentially visit websites from $W_p$. After visiting the last website on the list, we save the browser profile. This profile, denoted as $P_p$, represents a simulated persona with an interest in the specific topic. This is **Step 1 (A1, A\*1, B1)** in Figure 2. We present various fingerprinting configurations in the browser extension. When the framework initializes, it will either load the spoofed fingerprints or not, depending on the specific settings. This ensures the framework can operate automatically under different configurations. All settings will activate simultaneously. It should be noted that the websites in **Step 1** are visited sequentially. **Step 1** and **Step 2** for different configurations **A1, A\*1, B1, A2, A\*2, B2** are run in parallel. The websites in **Step 1** are listed in Appendix A.1. The websites in **Step 2** are listed in Appendix A.2

### 4.2 Collecting Bids and HTTP Data

To collect bids data and investigate factors affecting bid behavior under the same browser profile, we first establish a data collection website list, $W_{bids}$. We create a detector to visit the Alexa top 10,000 websites list. Functions are developed to identify websites utilizing the Header Bidding implementation tool Prebid.js. The default API, *pbjs*, facilitated by Prebid.js, furnishes bidding data from various advertisers. Upon detection of Prebid.js during a website visit and successful retrieval of bidding records via the *pbjs* API, the website is

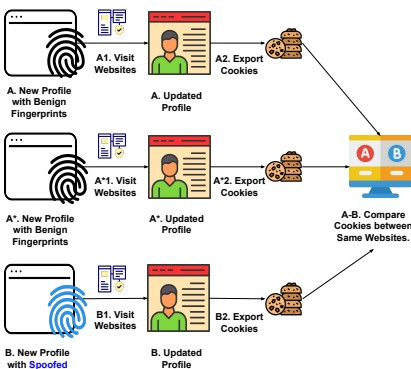

*Figure 3: High level overview of cookie restoration experiment. In **Step 1**, FPTrace will control the browser to visit websites. In **Step 2**, FPTrace will export all cookies including including 1st party and 3rd party cookies. In **A** and **A\***, FPTrace will use the same browser fingerprints, and same experiment conditions. In **B**, FPTrace will use a different browser fingerprint. Then FPTrace will do cookie comparisons between **A**, **A\*** and **B**.*

flagged. Subsequently, all flagged websites are stored in the list $W_{bids}$. After assembling the list, we use the $P_p$ profile to visit $W_{bids}$. **Step 2 (A2, A\*2, B2)** in Figure 2 represents the process. After we gathering bid data and HTTP data from the **Step 3 (A3, A\*3, B3)** in Figure 2, we can do the analysis to measure the targeting. The updated profile will remain the same during **Step 2** and **Step 3**. This could eliminate of the effects from the previous visited websites in **Step 2**. In **Step 3 (A3, A\*3, B3)** depicted in Figure 2, it is important to highlight that we extend our consideration beyond just the winning bid to encompass all advertising participants. Even if these participants did not secure the bid, their bids still offer valuable insights into their interest in the current visitor. Therefore, we carefully collect and consider their bids as part of our methodology.

To explore the impact of browser fingerprinting on bid changes, we first use the true browser fingerprints in experiments **A** and **A\***, which are represented in Figure 2. It should be noted that the updated profiles in A and A\* are different. We do not use the same profile for the different experiment, even the settings are all the same. We also have another experiment, which is denoted as **B** in Figure 2. In this setting, we replace the true browser fingerprint by using a brand new spoofed fingerprint.

### 4.3 Detecting Cookie Restoration

Our next step is to examine cookie restoration under various settings. In the "True Setting" ($S_{true}$), which is represented as **A** in Figure 3, we initiate a new browser profile without any pre-existing data, including browsing history and cookies. By using the OpenWPM framework to load the browser profile, and sequentially visit a list of websites, the profile is updated after each visit. The resulting profile is termed $P_{true}$, and the exported cookie from profile $P_{true}$ is designated as $C_{true}$.

Following this, we erase all data, including browsing history and cookies, from the browser profile. Initially, we unzip the browser profile, remove all data associated with history and cookies, and subsequently zip the remaining files to generate the new browser profile. To ensure thorough data removal, we opt for creating a new

*Table 1: Various configurations involving browser profiles, cookies, browser fingerprinting, and IP addresses.*

| Setting Number | New Profile? | Have Cookies? | True Fingerprint? | True IP? |
|---|---|---|---|---|
| a | Yes | No | Yes | Yes |
| b | No | Yes | Yes | Yes |
| c | No | Yes | No | Yes |
| d | No | No | Yes | Yes |
| e | No | No | No | Yes |

browser profile instead of using $P_{true}$. This new profile, labeled $P_0$, is employed to visit the same websites as in $S_{true}$ while operating on the same machine and IP address. The exported cookie from $P_0$ is denoted as $C_0$, and this setting is identified as $S_0$, which is represented as **A\*** in Figure 3.

To detect cookie restoration, we compare all cookies between $C_{true}$ and $C_0$, matching them based on cookie key name, value, and owner. Cookies in $C_0$ that precisely match those in $C_{true}$ with regard to key name, value, and owner are considered restored. We collect these restored cookies into a new set called $C_{True-0}$.

### 4.4 Detecting Fingerprinting Related Cookie Restoration

Cookie restoration may result from factors like supercookies and IP addresses. We explore whether browser fingerprinting also plays a role in cookie restoration. In setting $S_1$, we maintain the same machine and IP address as in $S_{true}$ but focus on the browser fingerprinting aspect. We integrate Gummy Browser spoofing techniques into an extension to manipulate browser fingerprinting.

After loading the Gummy Browser extension and starting with a fresh browser profile, we visit the same website list as in $S_{true}$. This process generates an updated browser profile, $P_1$, and the associated cookie, $C_1$. This is represented as **B** in Figure 3.

Distinguishing setting $S_1$ from $S_{true}$, the only variable altered is the browser fingerprinting. Therefore, any cookies in $C_{True-0}$ that cannot be found in $C_1$ serve as evidence that browser fingerprinting contributes to cookie restoration. These cookies are not fully restored when browser fingerprinting is modified.

Additionally, when comparing $C_{True-0}$ and $C_1$, we may identify cookies with altered values but matching key names and owners. This observation further confirms that browser fingerprinting influences cookie restoration by demonstrating that cookie values change when fingerprinting is manipulated.

It should be noted that all settings **A1, A\*1, B1** will activate simultaneously. The websites in **Step 1** are listed in Appendix A.2

## 5 EXPERIMENT SETTING

In this section, the objective of our experiment is to investigate the utilization of fingerprinting techniques in online advertising, particularly focusing on its impact on bid behavior and cookie restoration. The experiment is structured into several steps, each designed to simulate different scenarios involving fingerprinting, cookies, and IP addresses.

### 5.1 Initial Profiling and Data Collection

The first step involves the creation of an initial profile by visiting a range of websites related to *"Computers"*. The full website list of $W_p$ here is listed in Appendix A.1. During this phase, cookies are enabled, and various fingerprinting techniques are deployed to

collect data for analysis. The recorded data includes fingerprinters used and data sharing activities. The generated profile is then saved for subsequent comparisons.

## 5.2 Variation in Scenarios

The second step introduces variations to the initial profile, encompassing different combinations of cookies, fingerprinting techniques, and IP addresses. The scenarios examined are as follows:

a. A new profile is introduced, and bids and cookies are collected while visiting websites. This scenario provides a baseline for comparison with true fingerprints and true IP addresses (new profile, true fingerprint, true IP).

b. Profile with cookies is employed, bids and cookies are collected. This scenario explores the impact of true fingerprinting with the same IP addresses. This scenario serves as a baseline for true fingerprinting (have cookies, true fingerprint, true IP).

c. Profile with cookies is employed, with a different IP address and using the alternate fingerprints, while collecting bids and cookies. This scenario delves into the effects of both fake fingerprints and fake IP addresses (have cookies, fake fingerprint, true IP).

d. Profile without cookies is used to visit websites, allowing the collection of bids and cookies (no cookies, true fingerprint, true IP).

e. Profile without cookies is used, but with a different fingerprints, and bids and cookies are collected. This scenario investigates the role of different fingerprints with a true IP address (no cookies, fake fingerprint, true IP).

We summarize the settings in Table 1.

## 5.3 Experiment Locations

FPTrace was primarily implemented on devices situated in the United States, where we gathered bid data, monitored HTTP events, and conducted cookie restoration detection. In addition to examining regions lacking explicit privacy regulations in the United States, we also evaluated the use of browser fingerprinting in areas where privacy regulations are in place. The location selection of GDPR and CCPA experiments are explained in Appendix A.4.

## 6 RESULTS AND EVALUATION

In this section, we first present the results of our experiments conducted in the United States, where privacy regulation protection was absent. These experiments were focused on data sharing and cookie restoration through browser fingerprinting, which fall outside the scope of current privacy regulations. Following this, we examine whether data sharing based on browser fingerprinting also takes place under the GDPR and CCPA privacy regulations in Appendix A.5 and A.6.

## 6.1 Bid Values and HTTP Events

*6.1.1 Have Cookies.* We collect and analyze bids data across different settings, specifically transitioning from "have cookies, have data, true fingerprints, true IP address" to "have cookies, have data, fake fingerprints, true IP address." The sole differentiating factor here is the presence of "true/fake fingerprints." To ensure data consistency, we conduct the experiment setting "have cookies, have data, true

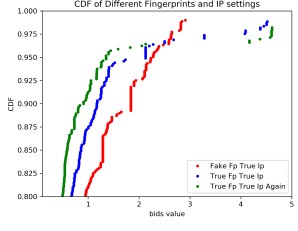 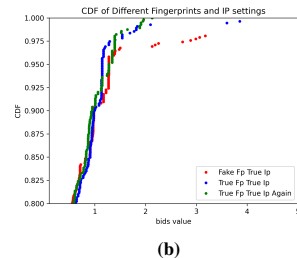

(a)                                        (b)

**Figure 4:** *Figure a is the CDF of different fingerprints and IPs settings in keeping cookies. Figrue b is the CDF of different fingerprints and IPs settings in removing cookies. The range of bids value is from 0 to 5. The range of CDF score is between 0.8 to 1. In Figure a, We can observe that the red curve is much different from the other two curves which are closer to each other, thus showing that fingerprinting is being used for tracking. In Figure We can observe that the red curve is much different from the other two curves which are closer to each other, thus showing that fingerprinting is being used for tracking.*

fingerprints, true IP address" twice to verify minimal bid data and HTTP data fluctuations. Table 2 illustrates that bid values between two instances of "True FP True IP" are quite similar, meeting our objective of running them twice. However, when comparing "True FP True IP" to "Fake FP True IP," in Table 2, there are substantial differences in median and maximum bid values. This suggests that changes in browser fingerprinting impact bid values, implying that alterations in browser fingerprinting influence targeting and tracking in advertising.

**Table 2:** *The bid value in different fingerprints and IPs settings. **Avg** represents the average of all bid value. **Median** represents the median bid value. **Min** represents the minimum bid value. **Max** represents the maximum bid value. We can observe that the first row values are different from the other two rows, which are similar to each other, thus showing that the change in FP creates a more marked impact on the bid values indicating that fingerprinting is being used for tracking.*

|  | Avg | Std | Median | Min | Max |
|---|---|---|---|---|---|
| **Fake FP True IP** | 0.60 | 1.14 | 0.19 | 0.00 | 10.02 |
| **True FP True IP** | 0.51 | 0.86 | 0.25 | 0.00 | 5.45 |
| **True FP True IP 2** | 0.46 | 0.93 | 0.23 | 0.00 | 5.52 |

We have also generated a CDF plot (see Figure 4a) using bid data from the two experiment settings. The distributions of CDF between different settings are similar when the CDF score is lower than 0.8, as all bids values are lower than 1. So we only display the distributions when the CDF scroe is larger than 0.8. In this CDF plot, the trends for two instances of "True FP True IP" are also similar, signifying that bid data remains stable under the same browser fingerprint. However, the introduction of fake fingerprints alters the trend of bid data. This indicates a significant difference in the distribution of bids between the two settings. The CDF plot provides evidence that changes in browser fingerprinting affect targeting and tracking in advertising.

Furthermore, we calculated the number of HTTP chains, syncing events, and total HTTP records using HTTP data from the two settings. The results are outlined in Table 3. It is evident that the number of total HTTP chains in two instances of "True FP True IP" is similar and exceeds 36,000. However, changing the browser

**Table 3:** *The number of events and data through HTTP data.* **Chains** *represents the number of chains we built using collected HTTP data.* **Syncing** *represents the number of chains containing the data sharing.* **Records** *represents the total number of HTTP records collected in this settings. We can observe that the first row values are different from the other two rows, which are similar to each other, thus showing that the change in FP creates a more marked impact on the bid values indicating that fingerprinting is being used for tracking.*

|                     | Chains | Syncing | Records |
|---------------------|--------|---------|---------|
| **Fake FP True IP** | 6345   | 421     | 109802  |
| **True FP True IP** | 36446  | 888     | 313094  |
| **True FP True IP 2** | 37929 | 895    | 318208  |

**Table 4:** *The bid value in different fingerprints and IPs settings.* **Avg** *represents the average of all bid value.* **Median** *represents the median bid value.* **Min** *represents the minimum bid value.* **Max** *represents the maximum bid value. Setting "True FP True IP 2" do keep the cookies. Other two settings do not keep the cookies.*

|                     | Avg  | Std  | Median | Min  | Max  |
|---------------------|------|------|--------|------|------|
| **Fake FP True IP** | 0.42 | 0.89 | 0.13   | 0.00 | 5.92 |
| **True FP True IP** | 0.38 | 0.54 | 0.19   | 0.00 | 5.80 |
| **True FP True IP 2** | 0.35 | 0.41 | 0.17  | 0.00 | 2.13 |

fingerprint reduces this number to 6,345, a substantial drop. This reduction indicates fewer HTTP chains can be established, implying a decrease in redirects and a reduced ability for websites to recognize the browser profile. A similar pattern emerges in syncing events: the number of syncing events in "Fake FP True IP" is 50% less than in "True FP True IP," signifying a 50% reduction in data sharing. The total number of HTTP records in "Fake FP True IP" is nearly 65% less than in "True FP True IP." The decrease in HTTP records also implies reduced recognition on the internet.

> **Takeaway [Have Cookies]:**
> (1) Bids trend and range are changed after changing the browser fingerprinting.
> (2) HTTP chains, syncing events and total records drop after changing the browser fingerprinting.

*6.1.2 No Cookies.* We gather and scrutinize bid data across various scenarios, particularly transitioning from a state characterized by "no cookies, have data, true fingerprints, true IP address" to a state with "no cookies, have data, true fingerprints, true IP address". To ensure the consistency of the data, we repeat the experiment under the setting "have cookies, have data, true fingerprints, true IP address" to confirm minimal variations in bid data and HTTP data. In the context of "true fingerprints", we introduce one fabricated fingerprint, while "fake fingerprints" includes another counterfeit fingerprint. Table 4 provides an overview of bid values across these three different configurations. When we keep the fingerprints consistent and eliminate cookies, the average and median bid values experience marginal increases, but the maximum bid value rises significantly. Following the alteration of fingerprints, both the average and maximum bid values increase, while the median value experiences a notable decrease. This suggests that changes in browser fingerprinting have an impact on bid values, underscoring their influence on targeting and tracking in advertising.

**Table 5:** *The number of events and data through HTTP data.* **Chains** *represents the number of chains we built using collected HTTP data.* **Syncing** *represents the number of chains containing the data sharing.* **Records** *represents the total number of HTTP records collected in this settings.* **CR** *represents the setting of Cookie Removed. All data are from settings including removing cookies.*

|                          | Chains | Syncing | Records |
|--------------------------|--------|---------|---------|
| **Fake FP True IP (CR)** | 9606   | 632     | 119673  |
| **True FP True IP (CR)** | 9796   | 624     | 124543  |
| **True FP True IP 2**    | 11242  | 649     | 128781  |

We have also generated a CDF plot (see Figure 4b) using bid data from the two experimental configurations. The distributions of CDF between different settings are similar when the CDF score is lower than 0.8, as all bids values are lower than 1. So we only display the distributions when the CDF scroe is larger than 0.8. In this CDF plot, the patterns for two instances of "True FP True IP" are notably similar, indicating that bid data remains stable when the same browser fingerprint is used. However, the introduction of fake fingerprints disrupts the trend of bid data, signifying a substantial difference in bid distribution between the two settings. The CDF plot furnishes evidence that variations in browser fingerprinting influence targeting and tracking in advertising.

Furthermore, we computed the number of HTTP chains, synchronization events, and total HTTP records using HTTP data from the two configurations. The outcomes are summarized in Table 5. It is apparent that the total number of HTTP chains remains comparable across all three settings. Upon the removal of cookies, this number decreased by approximately 13 percent, and further dropped by 2 percent when cookies were changed. The number of synchronization events exhibited consistency across the three settings, while the total count of HTTP records continued to decrease after cookies were removed and fingerprints were altered. The decrease in HTTP records also suggests reduced online recognition.

> **Takeaway [No Cookies]:**
> (1) Bids trend and range are changed after removing cookies and changing the browser fingerprinting.
> (2) HTTP chains, and total records drop after removing cookies and changing the browser fingerprinting.

Based on our analysis of bid values and HTTP events, we can draw the conclusion that browser fingerprinting indeed plays a significant role in targeting and tracking within the realm of advertising.

## 6.2 Cookie Restoration

After the cookie data comparison, we totally detected 90 cookie key and host pairs, for example: key "acc_segment" and host "sgqcvfjvr-.onet.pl". Fingerprinting related cookie restoration behaviors are found under all settings: All changes, Languages, OS, appVersion, UserAgent, Vendor, CookieEnabled, doNotTrack, Language, Plugin, Canvas. In UserAgent, Languages and Language settings, we not only change the value JS API "Navigator", but also change the value in HTTP header. The following is the example of cookie value changed after fingerprints are changed.

```
domain: google.it
key: OTZ
value:
```

```
1. 7151858_76_80_104160_76_446820(UA: chrome)
2. 7151859_76_80_104160_76_446820(UA: firefox)
```

In total, we documented 378 instances of cookie restoration related to fingerprinting across 90 unique combinations of cookie keys and host pairs across all settings. Subsequently, we conducted a comprehensive manual inspection of all 90 cookie key and host pair combinations. However, the results of our examination do not provide substantial support for the assertion that these cookie restorations are linked to fingerprinting.

Throughout our manual inspection, we observed that some values did indeed change when we altered the browser fingerprint, while others exhibited variations correlating with the passage of time. Additionally, certain values appeared to be random, deriving from an unidentified source. For identical cookie key and host pairs, we encountered instances where the same value persisted during two inspections with different fingerprints, as well as situations where the same value occurred during two inspections with the same fingerprints. Furthermore, some values incorporated fingerprinting feature values, suggesting that alterations in browser fingerprints influenced the cookie value. Nonetheless, these changes cannot be unequivocally attributed to browser fingerprinting restoration.

> **Takeaway [Cookie Restoration]:**
> (1) During the data collection process for automation, we observed that certain cookies are reinstated when fingerprints remain consistent. However, when these fingerprints are altered, the values of these cookies either change or they vanish.
> (2) Upon manual examination, the restoration of these cookies was determined not to be related to fingerprinting.

In light of our thorough manual inspection, we cannot definitively assert that browser fingerprinting is employed to restore cookies.

## 7 DISCUSSION AND LIMITATIONS

Our experiment was conducted using IP addresses from two locations in the United States, both of which are located in the United States and are not subject to privacy regulations such as GDPR [15] or CCPA [16]. In regions protected by such regulations, trackers like cookies are prohibited from tracking users once they opt out. However, our experiment has revealed that advertisers may employ browser fingerprinting to track users without providing any notification. It remains uncertain whether advertisers can continue using browser fingerprinting to track users, as there is currently no established framework for auditing advertisers in this context. It's important to note that our experiment cannot be utilized to assess advertisers' behavior within the constraints of privacy regulations.

Another limitation of our study is that all experiments were conducted on the Linux platform. We cannot determine whether users of Windows devices, MacOS devices, or mobile devices can still be tracked by advertisers using browser fingerprinting techniques. While some of our fake fingerprint data were obtained from Windows devices, MacOS devices, or mobile devices, which we used to emulate our experimental device browsers, it would be valuable to incorporate real Windows devices, MacOS devices, or mobile devices in the "True Fingerprints" settings to gain a more comprehensive understanding.

Additionally, there is uncertainty regarding whether websites visited by FPTrace can accurately distinguish between visits from a crawler and those from real users. Despite our efforts, such as altering JS API values and simulating human behaviors, we cannot be entirely certain that there are no undisclosed techniques for detecting bot visits. If FPTrace's visits are identified as originating from a bot, the accuracy of our results may be compromised.

Compared to the work of Liu et al.[33], there are notable differences in our study in both experimental design and research objectives. While our work focuses on exploring various fingerprinting settings and assessing whether different privacy regulations can constrain fingerprinting techniques, Liu et al.[33] did not involve any specific fingerprinting configurations. Instead, their research aimed to evaluate whether CMPs, websites, or advertisers comply with users' consent choices.

## 8 RELATED WORK

Previous research has examined the connection between online tracking and ad targeting. Wills et al.[44] explored Google and Facebook's ad systems, identifying various ad types. Google sometimes showed non-contextual ads related to sensitive topics, even without relevant user activity. On Facebook, external browsing had no clear link to ads, but using the "Like" feature on third-party content influenced ad recommendations. Zeng et al.[47] studied the impact of user attributes, demographics, and contextual factors on ad targeting and bid values. Using data from 286 participants across 10 websites, they found targeting was primarily shaped by the website, retargeting methods, and user behavior, with demographics playing a minor role. Variations in bid values were mostly linked to the website and user actions, highlighting the significance of contextual targeting. Cassel et al.[21] analyzed web tracking across devices, noting fewer tracking requests on mobile. Privacy-focused browsers reduced tracking but showed susceptibility to fingerprinting, without delving into its role in ads. Fouad et al.[26] investigated cookie restoration using VPNs, which might introduce latency. Our approach, using real IPs and manual inspections, revealed that factors like time and website data could lead to restoration resembling fingerprinting.

## 9 CONCLUSION

In this study, we address a gap in research and regulatory practices by investigating browser fingerprinting's role in ad tracking. We introduce "FPTrace", a framework for detecting fingerprinting usage, especially when changes in fingerprints disrupt ad targeting. FPTrace is evaluated in web environments where user data is routinely collected for ads. Our method involves A/B experiments, data leakage analysis, fingerprint manipulation, and examining ad bidding behavior. We explore how altered fingerprints affect cookie restoration, hypothesizing that prior user knowledge influences bid changes.

Key contributions include the FPTrace framework, integrated with OpenWPM, which simulates user interactions, collects bid data, and records HTTP data. Our findings show browser fingerprinting significantly impacts ad tracking, with noticeable differences in bid values and a reduction in HTTP records when fingerprints change. While we find some connection between fingerprints and cookies, evidence of direct involvement in cookie restoration is inconclusive.

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

# A APPENDIX

## A.1 Training Websites

(1) https://www.bestbuy.com/site/electronics/computers-pcs/abcat0500000.c?id=abcat0500000

(2) https://www.newegg.com/Computer-Systems/Store/ID-3

(3) https://www.bestbuy.com/site/searchpage.jsp?id=pcat17071&st=best+buy+computers+for+sale

(4) https://www.pcmag.com/picks/the-best-desktop-computers

(5) https://www.consumerreports.org/cro/computers/buying-guide/index.htm

(6) https://www.walmart.com/cp/computers/3951

(7) https://www.amazon.com/Computers-Tablets/b?ie=UTF8&node=13896617011

(8) http://www.newegg.com/

(9) https://www.dell.com/en-us

(10) https://www.hp.com/us-en/shop/cat/desktops

(11) https://www.adorama.com/l/Computers

(12) https://edu.gcfglobal.org/en/computerbasics/what-is-a-computer/1/

(13) https://www.tomsguide.com/best-picks/best-computers

(14) https://www.costco.com/computers.html

(15) https://www.microsoft.com/en-us/store/b/pc

(16) https://www.britannica.com/technology/computer

(17) https://www.cyberpowerpc.com/page/Intel/11th-Gen-Desktops/

(18) https://www.backmarket.com/refurbished-pc-desktop-computer.html

(19) https://www.bhphotovideo.com/c/browse/Computers-Solutions/ci/9581

(20) https://www.microcenter.com/

(21) https://www.target.com/c/computers-office-electronics/-/N-5xtfc

(22) https://www.officedepot.com/cm/tech/shop-pcs

(23) https://www.tigerdirect.com/

(24) https://www.staples.com/Laptops-Computers/cat_SC3

(25) https://www.lenovo.com/us/en/

(26) https://www.cdw.com/content/cdw/en/products/computers.html

(27) https://www.energystar.gov/products/computers

(28) https://www.xidax.com/xidaxsale

(29) https://buildredux.com

(30) https://www.corsair.com

(31) https://www.magicmicro.com

(32) https://www.thebudgetpcbuilder.com/

(33) https://www.ebay.com/b/Computers-Tablets-Network-Hardware/58058/bn_1865247

(34) https://computersfortheblind.org/

(35) https://www.nytimes.com/wirecutter/electronics/computers/

(36) https://www.samsung.com/us/computing/

(37) https://www.lg.com/us/computers

(38) https://www.apple.com/

(39) https://www.sciencedirect.com/journal/computers-environment-and-urban-systems

(40) https://www.ibuypower.com/

## A.2 Testing Websites

(1) https://tribunnews.com

(2) https://grid.id

(3) https://nytimes.com

(4) https://ettoday.net

(5) https://kompas.com

(6) https://indiatimes.com

(7) https://globo.com

(8)  https://liputan6.com

(9)  https://uol.com.br

(10)  https://speedtest.net

(11)  https://alwafd.news

(12)  https://theguardian.com

(13)  https://espn.com

(14)  https://cnet.com

(15)  https://brilio.net

(16)  https://businessinsider.com

(17)  https://foxnews.com

(18)  https://breitbart.com

(19)  https://metropoles.com

(20)  https://ladbible.com

(21)  https://investopedia.com

(22)  https://realtor.com

(23)  https://nypost.com

(24)  https://sportbible.com

(25)  https://hurriyet.com.tr

(26)  https://goo.ne.jp

(27)  https://softonic.com

## A.3  GDPR and CCPA Configuration

In GDPR and CCPA experiments, we set 6 different settings on each CMP and location combination. The number of each setting is described as following:

(0)  New profile, True fingerprint.
(1)  Trained profile, True fingerprint, Have Cookies. (Base Line)
(2)  Trained profile, Fake fingerprint, Have Cookies.
(3)  Trained profile, True fingerprint, Have Cookies. (Same setting as 1)
(4)  Trained profile, Fake fingerprint, No Cookies.
(5)  Trained profile, True fingerprint, No Cookies.

The complete workflow of GDPR and CCPA experiments is listed in Figure 5. Four CMPs Cookiebot, Quantcast, Onetrust, and Didomi CMPs are employed in our experiments, along with the central

opt-out platform NAI. True or fake fingerprints are set in step **C** in Figure 5.

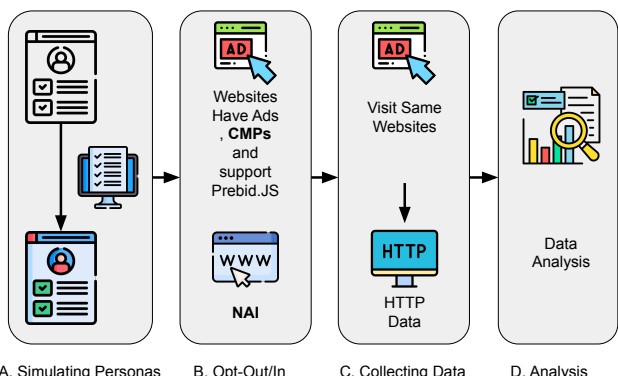

**Figure 5:** *High level overview of GDPR/CCPA experiment.* **Step A** *represents the training persona, with the original fingerprint.* **Step B** *involves Opt Out or Opt In actions on websites with ads, utilizing CMPs, and employing Prebid.js, or on the central Opt Out website NAI.* **Step C** *involves the collection of HTTP data.* **Step D** *represents the data analysis.*

In our experiments, "Opt out" refers to scenarios where the user declines to provide consent to the website, while "Opt in" denotes instances where the user grants consent. All interactions with these different CMPs are conducted automatically.

## A.4  Experiment Locations

Privacy regulations like the General Data Protection Regulation (GDPR) [15] and the California Consumer Privacy Act (CCPA) [16] are designed to safeguard user personal information. Under GDPR, customers have assurance against privacy breaches until they consent on websites. For CCPA protection, customers must actively refuse consent on websites to avoid privacy breaches. In our experimental setup, we initially evaluate data sharing related to fingerprinting and cookie restoration using local IP addresses in locations within the United States, where privacy regulation protection was absent. Subsequently, we implement FPTrace on AWS servers located in Frankfurt, Germany, and California, USA. Frankfurt-based visits fall under GDPR protection, while those from California are under CCPA. The aim of selecting these varied regions is to discern the impact of browser fingerprinting in environments with and without privacy regulation protection.

For visitors accessing websites from Europe or California, USA, the sites are expected to present a consent panel, allowing them to either agree or refuse consent. CMPs are third-party services that facilitate this consent process for users. There is a variety of CMP services available. As identified by Liu et al.[33], four prominent CMPs - Cookiebot, Quantcast, Onetrust, and Didomi - are among the most widely used and can be considered for use in this context. FPTrace can automatically handle the opt-out and opt-in. To ensure compliance with Didomi, we first verify the existence of the consent dialog using Didomi.notice.isVisible. We then employ Didomi.setUserDisagreeToAll to opt out and proceed to conceal the

consent dialog by adjusting the display attributes of its markup to 'none'. Similarly, for OneTrust, we confirm the presence of the consent dialog via window.OneTrust, executing window.OneTrust.RejectAll to opt out and subsequently conceal the dialog. In the case of CookieBot, we validate the presence of the consent dialog through window.Cookiebot, navigating the DOM to locate the opt-out button with the ID CybotCookiebotDialogBodyButtonDecline and activating it. Regarding Quantcast, we search the DOM for the consent dialog using the qc-cmp2-summary-buttons class name, clicking the button labeled 'Reject' or similar. If the rejection option is not initially visible, we expand the dialog by selecting the 'more options' button and then proceed to click 'Reject All'. Additionally, we assess the effectiveness of the central opt-out service Network Advertising Initiative (NAI) [10], which enables users to refuse consent through a single visit to its website. Consequently, other websites should comply by not utilizing the users' consent if this service is used.

## A.5 Browser Fingerprinting Data Sharing under Privacy Regulations

Tables of full results from CMP Cookiebot, Didomi and Quantcast, and NAI are listed in Appendix A.6.

Under GDPR regulations, Cookiebot showed a significant increase in the number of HTTP Chains, Syncing Events, and HTTP records when cookies were removed, observed in both Opt Out and Opt In settings. However, using a fake fingerprint did not result in notable changes in either setting. Full details are listed in Appendix A.6 Table 9.

With Didomi, also under GDPR, the various settings, including Opt Out versus Opt In and True/Fake fingerprint or Have/No Cookies scenarios, did not exhibit substantial differences. Full details are listed in Appendix A.6 Table 11.

In the case of Onetrust, under GDPR conditions, an increase in Syncing Events was observed when the system utilized fake fingerprints. This rise was particularly notable in scenarios combining Fake Fingerprint and No Cookies in the Opt Out setting, suggesting that browser fingerprinting might be used for data syncing even after users deny consent on websites using Onetrust. Full details are listed in Table 7.

For Quantcast, regardless of Opt Out or Opt In settings under GDPR, the application of Fake Fingerprint led to a more pronounced increase in syncing events than in HTTP chains. This suggests that browser fingerprinting might be employed for syncing data regardless of whether the user consents or denies consent on websites using Quantcast. Full details are listed in Appendix A.6 Table 13.

Lastly, in the context of NAI under GDPR, a pattern similar to Quantcast was observed. The number of syncing events and HTTP chains significantly increased when the fingerprints were altered, indicating that browser fingerprinting might be used for data syncing irrespective of the user's decision to give or reject consent on the NAI website. Full details are listed in Appendix A.6 Table 15.

> **Takeaway [Browser Fingerprinting in GDPR]:**
> (1) There is no conclusive evidence to suggest that **Cookiebot** and **Didomi** participate in data sharing through browser fingerprinting for user identification in both *Opt Out* and *Opt In* settings. Websites utilizing **Onetrust** might not be involved in data sharing activities that use browser fingerprinting to identify users in *Opt In* setting.
> (2) It appears that websites utilizing **Onetrust**, **Quantcast**, and **NAI** might be involved in data sharing activities that use browser fingerprinting to identify users in *Opt Out* setting. **Quantcast** and **NAI** might be involved in data sharing activities that use browser fingerprinting to identify users in *Opt In* setting.

Under the CCPA regulation, Cookiebot's diverse settings, such as Opt Out vs Opt In and True/Fake fingerprint or Have/No Cookies scenarios, showed no significant differences. Full details are listed in Appendix A.6 Table 8.

Likewise, under CCPA regulation, Didomi's settings, including Opt Out vs Opt In and True/Fake fingerprint or Have/No Cookies scenarios, displayed no notable differences. Full details are listed in Table Appendix A.6 10.

Under CCPA regulations, with Onetrust, a notable disparity was observed in the Opt Out setting between settings 2 and 1, as well as between 2 and 3. This suggests possible use of browser fingerprinting in data synchronization. Full details are listed in Table 6.

In the case of Quantcast, under CCPA regulation, the various settings including Opt Out vs Opt In and True/Fake fingerprint or Have/No Cookies scenarios, also showed negligible differences. Full details are listed in Appendix A.6 Table 12.

Lastly, under CCPA regulation with NAI, in the Opt Out setting, there was a significant difference between setting 2 and both settings 1 and 3, indicating a potential use of browser fingerprinting for data synchronization. Full details are listed in Appendix A.6 Table 14.

> **Takeaway [Browser Fingerprinting in CCPA]:**
> (1) There is no conclusive evidence to suggest that **Cookiebot**, **Didomi** and **Quantcast** participate in data sharing through browser fingerprinting for user identification in both *Opt Out* and *Opt In* settings. Websites utilizing **Onetrust**, and **NAI** might not be involved in data sharing activities that use browser fingerprinting to identify users in *Opt In* setting.
> (2) It appears that websites utilizing **Onetrust**, and **NAI** might be involved in data sharing activities that use browser fingerprinting to identify users in *Opt Out* setting.

## A.6 Analysis of HTTP data under GDPR and CCPA

Within this section, we provide information in additional tables for the analysis of HTTP data under the protection of GDPR or CCPA.

**Table 6:** *The number of events and data through HTTP data. The CMP is: Onetrust and the Privacy Regulation is: CCPA.* **Total HTTP Records** *represents the total number of HTTP records collected in this settings.* **Total HTTP Chains** *represents the number of chains we built using collected HTTP data.* **Syncing Events** *represents the number of chains containing the data sharing. All data are from settings including removing cookies. We can analyze the data from both setting 1 and setting 2. There are minimal differences observed in Opt In setting. A significant difference was noticed in the Opt Out setting when comparing settings 2 and 1, suggesting that fingerprint changes may affect the tracking.*

| | Onetrust | | | | | |
|---|---|---|---|---|---|---|
| | Opt Out | | | Opt In | | |
| | HTTP Chains | Syncing Events | HTTP Records | HTTP Chains | Syncing Events | HTTP Records |
| 0 | 1549 | 302 | 50928 | 1178 | 334 | 43107 |
| 1 | 724 | 199 | 38720 | 754 | 238 | 39001 |
| 2 | 2216 | 395 | 46595 | 1593 | 401 | 44221 |
| 3 | 1588 | 407 | 46257 | 1618 | 381 | 49725 |
| 4 | 1204 | 368 | 45948 | 1657 | 449 | 45742 |
| 5 | 1084 | 227 | 42639 | 1663 | 417 | 50911 |

**Table 7:** *The number of events and data through HTTP data. The CMP is: Onetrust and the Privacy Regulation is: GDPR.* **Total HTTP Records** *represents the total number of HTTP records collected in this settings.* **Total HTTP Chains** *represents the number of chains we built using collected HTTP data.* **Syncing Events** *represents the number of chains containing the data sharing. All data are from settings including removing cookies. We can analyze the data from both setting 1 and setting 2. There are minimal differences observed in Opt In setting. A significant difference of syncing events was noticed in the Opt Out setting when comparing settings 2 and 1, suggesting that fingerprint changes may affect the tracking.*

| | Onetrust | | | | | |
|---|---|---|---|---|---|---|
| | Opt Out | | | Opt In | | |
| | HTTP Chains | Syncing Events | HTTP Records | HTTP Chains | Syncing Events | HTTP Records |
| 0 | 400 | 10 | 38194 | 423 | 33 | 38587 |
| 1 | 400 | 21 | 37350 | 505 | 36 | 44067 |
| 2 | 455 | 42 | 38636 | 485 | 61 | 38783 |
| 3 | 474 | 29 | 39754 | 546 | 41 | 42426 |
| 4 | 708 | 99 | 42782 | 482 | 47 | 39095 |
| 5 | 516 | 34 | 41560 | 395 | 19 | 38807 |

**Table 8:** *The number of events and data through HTTP data. The CMP is: Cookiebot and the Privacy Regulation is: CCPA. Total HTTP Records represents the total number of HTTP records collected in this settings. Total HTTP Chains represents the number of chains we built using collected HTTP data. Syncing Events represents the number of chains containing the data sharing. All data are from settings including removing cookies. We can analyze the data from both setting 1 and setting 2. There are minimal differences observed in both Opt Out and Opt In settings.*

| | Cookiebot | | | | | |
| | Opt Out | | | Opt In | | |
| | HTTP Chains | Syncing Events | HTTP Records | HTTP Chains | Syncing Events | HTTP Records |
|---|---|---|---|---|---|---|
| 0 | 248 | 61 | 7934 | 315 | 104 | 8863 |
| 1 | 392 | 175 | 9197 | 459 | 125 | 9568 |
| 2 | 407 | 149 | 9249 | 509 | 154 | 9251 |
| 3 | 363 | 120 | 8981 | 344 | 94 | 8365 |
| 4 | 238 | 55 | 7727 | 505 | 141 | 9175 |
| 5 | 241 | 57 | 7664 | 292 | 131 | 7959 |

**Table 9:** *The number of events and data through HTTP data. The CMP is: Cookiebot and the Privacy Regulation is: GDPR. Total HTTP Records represents the total number of HTTP records collected in this settings. Total HTTP Chains represents the number of chains we built using collected HTTP data. Syncing Events represents the number of chains containing the data sharing. All data are from settings including removing cookies. We can analyze the data from both setting 1 and setting 2. There are minimal differences observed in both Opt Out and Opt In settings.*

| | Cookiebot | | | | | |
| | Opt Out | | | Opt In | | |
| | HTTP Chains | Syncing Events | HTTP Records | HTTP Chains | Syncing Events | HTTP Records |
|---|---|---|---|---|---|---|
| 0 | 133 | 5 | 7515 | 128 | 3 | 7068 |
| 1 | 118 | 3 | 7080 | 118 | 4 | 7226 |
| 2 | 123 | 4 | 7537 | 119 | 3 | 7116 |
| 3 | 123 | 5 | 7512 | 118 | 5 | 7371 |
| 4 | 253 | 14 | 15836 | 223 | 6 | 14367 |
| 5 | 228 | 8 | 14423 | 229 | 6 | 15352 |

**Table 10:** *The number of events and data through HTTP data. The CMP is: Didomi and the Privacy Regulation is: CCPA. Total HTTP Records represents the total number of HTTP records collected in this settings. Total HTTP Chains represents the number of chains we built using collected HTTP data. Syncing Events represents the number of chains containing the data sharing. All data are from settings including removing cookies. We can analyze the data from both setting 1 and setting 2. There are minimal differences observed in both Opt Out and Opt In settings.*

| | Didomi | | | | | |
| | Opt Out | | | Opt In | | |
| | HTTP Chains | Syncing Events | HTTP Records | HTTP Chains | Syncing Events | HTTP Records |
|---|---|---|---|---|---|---|
| 0 | 1954 | 295 | 136427 | 1718 | 255 | 130499 |
| 1 | 1862 | 235 | 128373 | 1933 | 267 | 132669 |
| 2 | 1554 | 271 | 130922 | 1858 | 245 | 125409 |
| 3 | 1760 | 234 | 127028 | 1491 | 244 | 132634 |
| 4 | 2329 | 312 | 139056 | 2279 | 330 | 134956 |
| 5 | 1951 | 226 | 127350 | 2042 | 305 | 134851 |

**Table 11:** *The number of events and data through HTTP data. The CMP is: Didomi and the Privacy Regulation is: GDPR. Total HTTP Records represents the total number of HTTP records collected in this settings. Total HTTP Chains represents the number of chains we built using collected HTTP data. Syncing Events represents the number of chains containing the data sharing. All data are from settings including removing cookies. We can analyze the data from both setting 1 and setting 2. There are minimal differences observed in both Opt Out and Opt In settings.*

| | Didomi | | | | | |
| | Opt Out | | | Opt In | | |
| | HTTP Chains | Syncing Events | HTTP Records | HTTP Chains | Syncing Events | HTTP Records |
|---|---|---|---|---|---|---|
| 0 | 833 | 31 | 121457 | 937 | 42 | 123001 |
| 1 | 871 | 31 | 127088 | 737 | 14 | 122245 |
| 2 | 880 | 37 | 130116 | 838 | 25 | 125372 |
| 3 | 753 | 19 | 121093 | 783 | 21 | 125504 |
| 4 | 899 | 38 | 132361 | 865 | 35 | 129829 |
| 5 | 841 | 46 | 122519 | 837 | 19 | 126583 |

**Table 12:** *The number of events and data through HTTP data. The CMP is: Quantcast and the Privacy Regulation is: CCPA. Total HTTP Records represents the total number of HTTP records collected in this settings. Total HTTP Chains represents the number of chains we built using collected HTTP data. Syncing Events represents the number of chains containing the data sharing. All data are from settings including removing cookies. We can analyze the data from both setting 1 and setting 2. There are minimal differences observed in both Opt Out and Opt In settings.*

| | Quantcast | | | | | |
| | Opt Out | | | Opt In | | |
| | HTTP Chains | Syncing Events | HTTP Records | HTTP Chains | Syncing Events | HTTP Records |
|---|---|---|---|---|---|---|
| 0 | 1862 | 302 | 71655 | 2252 | 319 | 72382 |
| 1 | 2132 | 346 | 75336 | 1409 | 192 | 68125 |
| 2 | 2050 | 343 | 71439 | 1674 | 288 | 70305 |
| 3 | 2040 | 362 | 74835 | 1767 | 299 | 69467 |
| 4 | 1751 | 281 | 68387 | 1740 | 268 | 67718 |
| 5 | 2136 | 343 | 71491 | 1995 | 287 | 71945 |

**Table 13:** *The number of events and data through HTTP data. The CMP is: Quantcast and the Privacy Regulation is: GDPR. Total HTTP Records represents the total number of HTTP records collected in this settings. Total HTTP Chains represents the number of chains we built using collected HTTP data. Syncing Events represents the number of chains containing the data sharing. All data are from settings including removing cookies. A significant difference of syncing events was noticed in both Opt Out and Opt In settings when comparing settings 2 and 1, suggesting that fingerprint changes may affect the tracking.*

| | Quantcast | | | | | |
| | Opt Out | | | Opt In | | |
| | HTTP Chains | Syncing Events | HTTP Records | HTTP Chains | Syncing Events | HTTP Records |
|---|---|---|---|---|---|---|
| 0 | 1077 | 42 | 67150 | 1130 | 44 | 68512 |
| 1 | 1150 | 47 | 70839 | 1095 | 49 | 65670 |
| 2 | 1184 | 97 | 67300 | 1261 | 66 | 69702 |
| 3 | 1033 | 41 | 65206 | 994 | 38 | 65513 |
| 4 | 1118 | 50 | 65522 | 980 | 34 | 63184 |
| 5 | 1067 | 28 | 65243 | 1012 | 30 | 63984 |

**Table 14:** *The number of events and data through HTTP data. The CMP is: NAI and the Privacy Regulation is: CCPA. Total HTTP Records represents the total number of HTTP records collected in this settings. Total HTTP Chains represents the number of chains we built using collected HTTP data. Syncing Events represents the number of chains containing the data sharing. All data are from settings including removing cookies. We can analyze the data from both setting 1 and setting 2. There are minimal differences observed in Opt In setting. A significant difference was noticed in the Opt Out setting when comparing settings 2 and 1, suggesting that fingerprint changes may affect the tracking.*

| | NAI | | | | | |
| | Opt Out | | | Opt In | | |
| | HTTP Chains | Syncing Events | HTTP Records | HTTP Chains | Syncing Events | HTTP Records |
|---|---|---|---|---|---|---|
| 0 | 949 | 203 | 32213 | 484 | 93 | 26908 |
| 1 | 600 | 149 | 27918 | 913 | 234 | 30701 |
| 2 | 1985 | 414 | 36313 | 971 | 211 | 29159 |
| 3 | 744 | 241 | 29736 | 1102 | 296 | 35130 |
| 4 | 1459 | 253 | 32307 | 1083 | 232 | 30027 |
| 5 | 1640 | 394 | 35534 | 1404 | 313 | 36936 |

**Table 15:** *The number of events and data through HTTP data. The CMP is: NAI and the Privacy Regulation is: GDPR. Total HTTP Records represents the total number of HTTP records collected in this settings. Total HTTP Chains represents the number of chains we built using collected HTTP data. Syncing Events represents the number of chains containing the data sharing. All data are from settings including removing cookies. A significant difference was noticed in both Opt Out and Opt In settings when comparing settings 2 and 1, suggesting that fingerprint changes may affect the tracking.*

| | NAI | | | | | |
| | Opt Out | | | Opt In | | |
| | HTTP Chains | Syncing Events | HTTP Records | HTTP Chains | Syncing Events | HTTP Records |
|---|---|---|---|---|---|---|
| 0 | 153 | 27 | 26855 | 378 | 51 | 34869 |
| 1 | 159 | 27 | 28697 | 202 | 33 | 31542 |
| 2 | 371 | 132 | 30118 | 420 | 96 | 30925 |
| 3 | 255 | 34 | 34368 | 173 | 40 | 29926 |
| 4 | 168 | 18 | 25830 | 305 | 54 | 29436 |
| 5 | 152 | 37 | 25767 | 140 | 16 | 26178 |

