# OpenReview forum: "The First Early Evidence of the Use of Browser Fingerprinting for Online Tracking"
_ACM.org/TheWebConf/2025/Conference — WWW 2025 Poster_

### Official Review · Reviewer_MxTL · 2024-11-11

**Novelty:** 5
**Technical Quality:** 5

**Review:**

The provided text presents several strengths that contribute to its effectiveness in discussing browser fingerprinting and its implications for online advertising. First, it identifies a significant gap in existing research, highlighting the need for deeper investigation into how browser fingerprinting is used for user tracking and targeted advertising. This establishes the relevance of the study and underscores its importance in the current digital landscape. The introduction of the FPTrace framework is a notable strength, as it offers a structured and innovative approach to analyzing the impact of fingerprinting on advertising practices. By emulating genuine user interactions and capturing detailed data, the study employs a rigorous methodology that enhances the reliability of its findings. Furthermore, the text effectively communicates the results, providing compelling evidence of the relationship between browser fingerprinting and advertisement tracking, along with the implications of privacy regulations like GDPR and CCPA. This thorough examination prompts critical discussions about user privacy and data security, making the research particularly timely and relevant. Overall, the text is well-structured and presents a clear narrative that emphasizes the significance of the study within the broader context of digital advertising and user privacy concerns.

Weakness

1. Although the abstract mentions “robust evidence” and “significant disparities,” it lacks specific quantitative results or key statistics. Including one or two figures could strengthen the claims made.
2. Page 1 - Introduction
2.1) The claim of being the "first study" to measure browser fingerprinting for user tracking could be challenged. If there are prior studies that address related issues or if previous research has laid the groundwork, this assertion may need further substantiation or nuance.
2.2) While introducing FPTrace is a significant contribution, the description may suggest complexity that could hinder practical application. If the framework is difficult to implement or requires extensive technical knowledge, its utility may be limited for some researchers or practitioners.
2.3) The acknowledgment that there is “no conclusive evidence” supporting the direct involvement of fingerprinting in cookie restoration raises questions about the overall robustness of the findings. It suggests a limitation in the study that might detract from the overall validity of the conclusions.
2.4) While discussing regulatory frameworks, the contributions could benefit from a deeper analysis of how fingerprinting practices might circumvent these regulations. Without this context, the implications of the findings might be less impactful.
3)Framework Implementation
3.1) Minor - partipated should be participated (3.2.1)
3.2) The text mentions several external tools (e.g., Gummy Browser, ModHeader) but does not provide enough information about their capabilities or limitations. This reliance may raise questions about the robustness and reliability of the spoofing methods used.

**Questions:**

Although OpenWPM includes features to simulate human-like browsing behavior, accurately mimicking all aspects of human interaction (e.g., decision-making, varying click patterns) can be challenging, potentially affecting the realism of the data collected. Is there any plan to handle it in future?

**Reviewer Confidence:**

4: The reviewer is certain that the evaluation is correct and very familiar with the relevant literature

**Scope:**

3: The work is somewhat relevant to the Web and to the track, and is of narrow interest to a sub-community

---

### Official Review · Reviewer_LfLB · 2024-11-21

**Novelty:** 5
**Technical Quality:** 3

**Review:**

This paper investigates an intriguing research question: determining whether browser fingerprints are used for online tracking. To address this question, the paper proposes examining whether changes in browser fingerprints influence advertising behaviors, such as bidding patterns and HTTP records. Furthermore, the authors present a framework to simulate user behavior and analyze the relationship between browser fingerprints, bidding activity, and cookies. The strengths and weaknesses of this paper are summarized below.

Strengths:

S1: The research question explored in this paper is novel and practical, which constitutes the paper's primary strength.

S2: The paper is generally well-structured and easy to follow.

Weaknesses/Questions:

W1: The primary concern is the reproducibility of the results. Based on the technical description, the authors rely on numerous tools and APIs that interact with online websites through complex pipelines, making reproducibility challenging. If the authors could curate a dataset for this promising task, the paper’s impact would be significantly enhanced.

W2: Simulating user behavior is a highly complex task. The paper does not sufficiently discuss how the proposed method simulates user behavior or evaluate its effectiveness. If the simulation is not realistic, the results may be less convincing.

W3: The paper does not include a comparison with any baselines, making it difficult to assess the effectiveness of the proposed framework.

**Questions:**

Please addresses the questions and concerns raised in the Weaknesses above.

**Reviewer Confidence:**

2: The reviewer is willing to defend the evaluation, but it is likely that the reviewer did not understand parts of the paper

**Scope:**

3: The work is somewhat relevant to the Web and to the track, and is of narrow interest to a sub-community

---

### Official Review · Reviewer_wwgK · 2024-11-29

**Novelty:** 4
**Technical Quality:** 4

**Review:**

Summary:
This paper presents a comprehensive study on the privacy-invasive use of browser fingerprinting for online tracking and targeted advertising. The authors introduce a novel framework designed to detect changes in online advertisements following alterations in browser fingerprinting settings.

Strengths:
1.This paper introduces a new framework FPTrace, which simulates user interactions, captures advertiser bids, records HTTP data, and manages cookies to measure the impact of browser fingerprints on ad tracking and targeting.
2.This paper 	provide evidence that browser fingerprinting is indeed used for advertisement tracking and targeting, as indicated by significant differences in bid values and a reduction in HTTP records after changes in fingerprinting.
3. The research further investigates the potential malicious use of fingerprinting under data protection regulations like GDPR and CCPA, revealing significant variations in HTTP chains and syncing events when browser fingerprints are altered, suggesting that fingerprinting might be used for user identification despite privacy regulations.

**Questions:**

1. The article describes the phenomenon of browser fingerprinting being potentially used for privacy invasion and introduces the objective of developing the FPTrace framework. However, the current motivation focuses mainly on the existence of the phenomenon and lacks a deeper discussion. For example, why existing methods are insufficient to address the issue, necessitating the introduction of a new framework.
2. The experiments, designed using FPTrace, are generally consistent with the research objectives. However, some parts of the methodology, such as cookie restoration detection and bid value analysis, require more explanation to clarify how they directly validate or disprove the research hypotheses.

**Reviewer Confidence:**

3: The reviewer is confident but not certain that the evaluation is correct

**Scope:**

3: The work is somewhat relevant to the Web and to the track, and is of narrow interest to a sub-community

---

### Official Review · Reviewer_2KpN · 2024-12-01

**Novelty:** 6
**Technical Quality:** 5

**Review:**

This is a very interesting and promising topic addressed by the authors on measuring whether browser fingerprinting is being used for the privacy-invasive purposes. The paper provided a structured method for investigating browser fingerprinting in a privacy-invasive context and evidence that browser fingerprinting is indeed utilized in advertisement tracking and targeting.

**Questions:**

Sorry, I am unable to provide any suggestions for revisions to this paper because I am not a professional in the relevant field.

**Reviewer Confidence:**

1: The reviewer's evaluation is an educated guess

**Scope:**

3: The work is somewhat relevant to the Web and to the track, and is of narrow interest to a sub-community

---

### Official Review · Reviewer_AMTJ · 2024-12-03

**Novelty:** 4
**Technical Quality:** 5

**Review:**

## **1. Quality**
The technical rigor of the paper is strong, with a well-designed methodology and comprehensive experiments. FPTrace, the proposed framework, integrates various tools and approaches to automate data collection and assess browser fingerprinting’s role in tracking and targeted advertising. However, there are areas where the depth of analysis could be enhanced.

### **Strengths**
- The integration of **OpenWPM**, **Prebid.js**, and **Gummy Browser** demonstrates a solid engineering effort.
- Quantitative analysis using metrics like **bid distributions**, **HTTP chain counts**, and **syncing events** is well-executed and supports the claims.
- The experiments are methodologically sound, employing controlled A/B testing to ensure reliability.

### **Weaknesses**
- The paper’s conclusions on **cookie restoration** are inconclusive, with no definitive evidence linking fingerprinting to restoration.
- The scope of experiments is limited to Linux platforms, which restricts generalizability.

---

## **2. Clarity**
The paper is generally clear, with a logical structure and detailed descriptions of the experiments. However, certain sections are dense and could benefit from simplification or restructuring for broader accessibility.

### **Strengths**
- Figures (e.g., methodology and CDF plots) effectively complement the text, making complex processes easier to understand.
- The introduction clearly frames the research question, emphasizing the gap in existing literature.

### **Weaknesses**
- Overly technical sections (e.g., cookie restoration analysis) may be difficult for readers outside the domain to follow.
- Acronyms and specialized terms (e.g., CMPs, Prebid.js) are not consistently defined at their first appearance.

---

## **3. Originality**
The paper makes a novel contribution by linking browser fingerprinting to ad tracking and targeting through a comprehensive measurement study. The introduction of FPTrace and its applications in this domain represent an advancement over prior tools.

### **Strengths**
- FPTrace is a unique contribution, combining multiple tools to explore the privacy-invasive use of fingerprinting in the ad ecosystem.
- The study fills a gap in research by going beyond the mere detection of fingerprinting scripts to assess their implications for tracking.

### **Weaknesses**
- While the experiments are innovative, some findings (e.g., fingerprinting's impact on ad bids) align with intuitions previously suggested in the literature, which could dilute the perception of novelty.

---

## **4. Significance**
This work addresses a critical issue in the online advertising ecosystem: the use of browser fingerprinting for tracking despite user consent under privacy regulations like GDPR and CCPA. The findings have implications for researchers, policymakers, and privacy advocates.

### **Strengths**
- The results highlight limitations in privacy regulations and suggest the need for stronger enforcement mechanisms.
- FPTrace has potential applications in regulatory compliance audits and privacy-enhancing technologies.

### **Weaknesses**
- The paper does not sufficiently explore the broader implications of its findings, such as how regulators or ad platforms might address the identified issues.
- The real-world scalability and adaptability of FPTrace remain untested.

---

## **Pros and Cons**

### **Pros**
1. **Innovative Framework**: FPTrace provides a new way to analyze browser fingerprinting in tracking and advertising.
2. **Comprehensive Experiments**: Includes A/B testing, bid analysis, and cookie restoration experiments under multiple configurations.
3. **Quantitative Evidence**: Presents robust metrics, such as bid value distributions and HTTP chain counts, to support its claims.
4. **Relevance to Privacy Regulations**: Highlights challenges in enforcing GDPR and CCPA in the context of fingerprinting.
5. **Methodological Clarity**: Logical flow of experiments with supporting visual aids and explanations.

### **Cons**
1. **Platform Bias**: Experiments are conducted exclusively on Linux, limiting the findings’ generalizability to other operating systems or devices.
2. **Inconclusive Findings**: No definitive evidence linking fingerprinting to cookie restoration reduces the impact of that claim.
3. **Limited Benchmarking**: Lack of direct comparison with existing tools (e.g., FPStalker, Liu et al.) makes it hard to contextualize FPTrace's novelty.
4. **Regulatory Analysis**: Insufficient discussion on the practical implications for policymakers or enforcement bodies.
5. **Complex Terminology**: Dense sections and inconsistent definitions of terms could hinder accessibility for a broader audience.

---

## **Conclusion**

The paper presents a valuable contribution to understanding browser fingerprinting in the context of ad tracking, with the innovative FPTrace framework being its most significant asset. While the technical quality is strong, gaps in benchmarking, platform diversity, and broader implications reduce the overall impact.

### **Suggestions for Improvement**
1. Benchmark FPTrace against similar tools to clarify its advancements.
2. Expand experiments to other platforms (e.g., macOS, Windows, mobile devices) for broader applicability.
3. Simplify dense sections and ensure consistent definition of terms and acronyms.
4. Include a discussion of ethical implications and actionable recommendations for policymakers and privacy advocates.
5. Test FPTrace in real-world, large-scale scenarios to demonstrate its scalability and practical relevance.

**Questions:**

## **Methodology and Experiment Design**
1. **Cross-Platform Applicability:**
   - Why were experiments limited to Linux platforms? Do you anticipate any differences in results when testing on Windows, macOS, or mobile devices?
   - Are there plans to extend FPTrace to support cross-platform experiments?

2. **Bot Detection in Experiments:**
   - How confident are you that the websites did not detect FPTrace as a bot?
   - Were any specific countermeasures implemented to ensure websites treated FPTrace visits as human interactions, and if so, how were they validated?

3. **Cookie Restoration Analysis:**
   - Could you clarify the manual inspection process for detecting fingerprinting-related cookie restoration?
   - Are there plans to refine this analysis or use automated tools to strengthen the claim?

---

## **Technical Contributions and Framework**
4. **Benchmarking FPTrace:**
   - How does FPTrace compare to existing tools like FPStalker or Liu et al.’s work in terms of feature coverage, accuracy, and scalability?
   - Could you provide quantitative benchmarks to highlight its advantages?

5. **Feature-Level Analysis of Fingerprinting:**
   - Did you investigate which specific fingerprinting features (e.g., canvas, user agent) are most impactful in influencing bid values or HTTP events?
   - Would feature-level results provide more actionable insights for regulators or privacy advocates?

6. **Scalability of FPTrace:**
   - Have you tested FPTrace in larger-scale environments, such as crawling thousands of websites simultaneously?
   - What are its performance limitations, and how might they be addressed?

---

## **Findings and Novelty**
7. **Statistical Robustness:**
   - Could you elaborate on the statistical tests used to validate the significance of changes in bid values and HTTP events across different fingerprinting configurations?
   - Are there confidence intervals or p-values associated with the findings?

8. **Impact of Privacy Regulations:**
   - How do you reconcile the observed use of fingerprinting under GDPR and CCPA with regulatory requirements for user consent?
   - Could this study inform specific actions that regulators or consent management platforms should take?

---

## **Broader Implications**
9. **Real-World Applicability:**
   - How feasible is FPTrace for use by regulators or consumer advocacy groups to audit compliance with GDPR and CCPA?
   - Are there any partnerships or initiatives planned to operationalize FPTrace in regulatory audits?

10. **Countermeasures by Advertisers:**
    - Have you observed any adaptive behavior from advertisers or tracking platforms in response to altered fingerprints?
    - What do you anticipate as the next steps for advertisers to evade detection, and how could FPTrace address them?

---

## **Ethical and Future Directions**
11. **Ethical Considerations:**
    - How do you address the ethical implications of revealing new methods to evade tracking regulations?
    - Could these findings inadvertently aid advertisers in refining fingerprinting methods?

12. **Future Enhancements to FPTrace:**
    - Do you plan to incorporate real-time monitoring or additional functionalities into FPTrace to make it more versatile for future studies?
    - Could FPTrace be extended to evaluate emerging tracking methods, such as cross-device or probabilistic tracking?

# **Ethical Issues in the Paper**

## **1. Ethical Implications of Exposing Tracking Methods**
   - **Issue:**
     The paper reveals methods that advertisers and tracking platforms could use to circumvent privacy regulations, such as GDPR and CCPA. By showcasing how browser fingerprinting can still be used to track users despite user opt-out, the research could unintentionally empower advertisers to refine their techniques for evading detection.
   - **Ethical Concern:**
     The authors must balance transparency and public awareness of privacy issues with the potential risk that malicious actors might exploit these findings to undermine user privacy further.

   - **Question for Authors:**
     How do you address the ethical concern that this paper could be used by advertisers to improve fingerprinting techniques and circumvent privacy laws?

---

## **2. User Consent and Privacy Violations**
   - **Issue:**
     The research demonstrates that fingerprinting can be used for targeted advertising even when users have opted out of tracking through mechanisms like cookie consent banners or privacy regulations (GDPR/CCPA). This implies that user consent mechanisms might be ineffective, which raises concerns about privacy violations.
   - **Ethical Concern:**
     The study underscores a flaw in the enforcement of user privacy, highlighting that regulations may not fully protect users' rights. This brings attention to the need for stronger protections, but it also emphasizes that current systems might allow for breaches of consent.

   - **Question for Authors:**
     Do you think your findings suggest the need for a redesign of consent management platforms, and if so, what improvements would you recommend?

---

## **3. Manipulation of Browsing Data**
   - **Issue:**
     The paper's experiments involve manipulating browser fingerprints (spoofing) to test the effect on ad bidding and tracking. While this is necessary for the research, there are ethical questions about how browser fingerprinting is used to collect and analyze user data.
   - **Ethical Concern:**
     Although the research does not directly collect personal user data without consent, it still involves analyzing and manipulating digital identities (fingerprints) that could be used to track individuals across websites. This raises the ethical concern of how much manipulation of online data is acceptable for research purposes.

   - **Question for Authors:**
     How do you ensure that the spoofing and manipulation of browser fingerprints in your experiments do not cross ethical boundaries, particularly in terms of respecting users' privacy and consent?

---

## **4. Transparency of Data Collection**
   - **Issue:**
     The study involves capturing data on ad bidding behavior and HTTP events from websites using FPTrace. While this is done for research purposes, it may raise questions about the transparency of data collection, particularly if any data is collected from real users without their explicit consent.
   - **Ethical Concern:**
     Even though the data collected is anonymized and used for research, the authors need to ensure that no personally identifiable information (PII) is inadvertently collected. Ethical data collection practices require informed consent, transparency, and clear communication to users about how their data might be used.

   - **Question for Authors:**
     Could you clarify the data collection process in your experiments? How do you ensure that no personally identifiable information (PII) is collected during your study?

---

## **5. Potential Harm to Users**
   - **Issue:**
     The paper illustrates how tracking through fingerprinting can continue even when users opt out, suggesting that fingerprinting poses significant risks to online privacy. However, there is no discussion about the potential harms this could cause to users, such as privacy violations or exposure to unwanted targeting.
   - **Ethical Concern:**
     While the research aims to expose a privacy issue, there is a need for the authors to consider the real-world impact on users and the broader societal implications of such widespread tracking and profiling.

   - **Question for Authors:**
     How do you assess the potential harm to users as a result of the widespread use of browser fingerprinting, and do you consider the findings of this research to have any direct implications for user protection?

---

## **6. Impact on Regulatory Frameworks**
   - **Issue:**
     The findings suggest that existing privacy regulations (GDPR/CCPA) may be ineffective in addressing the full scope of online tracking through fingerprinting. The research implies that fingerprinting can bypass opt-out mechanisms, raising concerns about the limitations of current regulatory frameworks.
   - **Ethical Concern:**
     While the paper identifies a gap in the regulations, it doesn't provide much discussion about the potential consequences of these findings on the future of digital privacy laws and enforcement. The paper should also address the ethical responsibility of researchers in guiding policy changes or improvements.

   - **Question for Authors:**
     How do you think your findings could inform future changes to privacy regulations, and what role should researchers play in advocating for stronger user protections?

---

# **Conclusion**
The paper raises important ethical issues related to online privacy, user consent, and the potential for misuse of research findings. While the authors aim to contribute positively by revealing privacy risks, these ethical concerns should be acknowledged and addressed to ensure responsible and transparent research practices.

**Reviewer Confidence:**

4: The reviewer is certain that the evaluation is correct and very familiar with the relevant literature

**Scope:**

3: The work is somewhat relevant to the Web and to the track, and is of narrow interest to a sub-community